# Enclosure in Combination with Mowing Simultaneously Promoted Grassland Biodiversity and Biomass Productivity

**DOI:** 10.3390/plants11152037

**Published:** 2022-08-04

**Authors:** Da Li, Yingying Nie, Lijun Xu, Liming Ye

**Affiliations:** 1Baicheng Institute of Animal Husbandry, Baicheng 137000, China; 2Institute of Agricultural Resources and Regional Planning, Chinese Academy of Agricultural Sciences, Beijing 100081, China; 3Department of Geology, Ghent University, 9000 Gent, Belgium

**Keywords:** meadow steppe, enclosure, grazing, degradation, conservation, forage, yield

## Abstract

Grassland is the primary land use in China, which has experienced extensive degradation in recent decades due to overexploitation. Here, we conducted field experiments to quantify the degraded grassland’s recovery rate in Northeast Inner Mongolia in response to restoration measures, including fallow + enclosure (FE) and mowing + enclosure (ME) in comparison to livestock grazing (LG), since 2005. Plant community properties were surveyed and aboveground biomass (AGB) sampled in summer 2013. Our results showed that the regional dominant species *Leymus chinensis* retained its dominance under FE, whereas a range of forb species gained dominance under LG. Vegetative cover was maximal under FE and minimal under LG. The least amount of vegetation development and AGB were observed under LG. However, plant diversity showed an opposite pattern, with maximal diversity under LG and minimal under FE. Statistical analysis revealed that AGB was negatively associated with plant diversity for all treatments except ME. For ME, a positive AGB-diversity relationship was characterized, suggesting that mowing intensity was a controlling factor for the AGB-diversity relationship. Overall, these results demonstrated that enclosure plus mowing represented an effective conservation measure that provided fair support to forage production and a progressive pathway to a more resilient grassland system.

## 1. Introduction

Grasslands are the primary land use in China, coving 40% of the country’s land mass [1], which is about three times that of croplands and two times that of forests. Half of the Chinese grasslands are located in the northern temperate zone, extending from the Northeast Plain in the east to Xinjiang in the west and forming an integral part of the Eurasian Steppe [2]. In aggregate, grasslands in China carry one-fifth of the sheep and one-eleventh of the cattle of the world and support the livelihood of some 400 million people in China alone [3]. Moreover, the northern grasslands provide a range of ecosystem services that have regional to national significance, including water conservation, climate stabilization, carbon sequestration, dust mitigation, etc. [1,4,5,6]. However, grasslands in China have experienced extensive degradation during the past few decades, mostly due to overexploitation [7,8,9]. Located in northeastern Inner Mongolia, the temperate meadow steppe in Hulunber represented one of the most conserved grasslands in the whole of China until recently, as characterized by its native vegetation, rich biodiversity, and diversified landscapes [10]. Grassland degradation in Hulunber expanded from 10% in extent in the 1980s to over 50% in the early 2000s, and is still developing at an average rate of 2% per year [11]. Degradation has not only led to decreases in grassland productivity and losses of biodiversity [12], but it has also disrupted ecosystem functioning and socioeconomic wellbeing [13,14].

Natural factors, such as climate and environmental changes, as well as human activities have been recognized as the major stresses to the grassland ecosystem [15,16,17,18]. As human societies continue to expand their environmental footprints, however, anthropogenic disturbances have outweighed the natural ones to become the major driver of grassland degradation [19,20]. The way in which humans utilize grasslands not only affects the individual grasslands, but also impacts the stability and resilience of the grassland ecosystem as a whole [21,22]. For example, grazing, mowing and enclosure are the three most commonly employed grassland management strategies in northern China. Increasing attention has been given in recent years to the impact that these grassland management regimes have on the diversity and other properties of the grassland community. Extensive studies have been carried out to investigate the effects of management measures involving, e.g., grassland type [23] and regional environment [24,25]. Studies have also been conducted on the attributes of the management practices, such as enclosure duration [26], grazing intensity [27] and mowing method [28]. However, systematic investigations that integrate multiple stresses using relatively long-term field experiments are still lacking. Here, we report experimental evidence on the effects of grazing, mowing and fence enclosure on the productivity and stability of grassland systems from a long-term field observation in Hulunber of Northeast Inner Mongolia in China. More specifically, the objectives of this paper are to: (1) analyze the effects of common conservation measures on plant community properties including species diversity and biomass productivity after five to eight years of continuous enforcement; (2) examine the biodiversity-productivity relationship to provide additional insights into the balance between grassland conservation and utilization; and (3) identify the most effective conservation measure or combination of measures to help build a more resilient grassland system.

## 2. Results

### 2.1. Plant Community Composition

Significant differences were observed in the composition of the plant communities under the influence of the five experimental treatments (Appendix A). In plot T1 under the fallow treatment, 15 plant species were identified across 14 genera and 9 families, whereas in plots T2 through T5 under their respective treatments, 28, 26, 12, and 43 species were identified. The *Leymus chinensis* species established dominance or remained dominant in plots T1 through T4. However, this dominance status was not observed in T5 under livestock grazing. The importance value (IV) of the species *Leymus chinensis* in T5 was measured as 0.04, which was significantly lower than, e.g., 0.68 in T1. Although the number of *Poaceae* species presented in plot T5 was 1.28–4.5 times more than in plots T1 through T4, the IVs of *Poaceae* species in T5 were substantially lower than in T1 through T4 by a margin of 57.4%, 16.3%, 54.0% and 58.3%, respectively (Figure 1). The forb species were dominant in plot T5 in terms of the number of species presented and their IVs.

### 2.2. Vegetation Coverage, Density and Growth

Significant differences were also observed in the vegetation coverage, density, litter accumulation and aboveground biomass (AGB) under the experimental treatments (Figure 2). The grassland under the enclosure protection (T1 and T2) developed a significantly higher level of plant cover than the grazed grassland (T5). Although plots T3 and T4 were also enclosure-protected, the plant cover in T3 and T4 was not significantly higher than in T5 because of the regular mowing disturbances. The grazed grassland not only had the lowest plant cover, but also had the lowest plant density, biomass production and litter accumulation. Plots that received the least disturbances (i.e., T1 and T2) produced the highest AGB compared to the plots that received regular mowing (T3 and T4) or constant grazing (T5). Plots T1 and T2 had relatively higher litter accumulation, whereas litter was largely removed in plots T3, T4 and T5 by means of biomass harvest or livestock grazing.

### 2.3. Plant Height and AGB of the Dominant Species

Significantly different levels of plant height were observed across plots T1 through T5 for the species *Leymus chinensis* (Figure 3). The average height of the *Leymus chinensis* species in the grassland under fallow (T1) was measured as 68.4 cm, which was 1.3-3.8 times higher than those in the other treatments. Moreover, *Leymus chinensis* species in plot T1 also had the highest AGB (183 g m^−2^), which was significantly higher than in plots T3, T4 and T5. Unsurprisingly, *Leymus chinensis* in plot T5 was measured as the lowest in both plant height and AGB, showing its nondominant status. It was interesting to observe that the difference between T3 and T4 was insignificant, suggesting that both plant height and AGB of the *Leymus chinensis* species were insensitive to mowing frequency.

### 2.4. Species Diversity

The plant species diversity tended to increase in the order of T1, T4, T3, T2 and T5 (Figure 4). The three categories of plant diversity indicators that we used, namely, the diversity, the evenness and the richness indices, showed a similar trend. The diversity level was at the lowest in the degraded grassland that had been placed in fallow and, in the meanwhile, protected by an enclosure (T1). In contrast, the highest diversity level was found in the livestock grazing grassland T5, showing a positive correlation between species diversity and the level of disturbances that the grassland received. Nevertheless, a significant difference was not observed between treatments T3 and T4, despite the mowing intensity difference between the two.

### 2.5. Relationship between AGB and Plant Diversity

Regression analyses showed that a negative trend existed between AGB and plant diversity. The characterized trends were consistently negative for all treatments except T3 (Figure 5). For T3, a positive trend was characterized, suggesting that an increase in the frequency of forage harvest from mowing once per three years (T4) to mowing once per year (T3) had triggered a direction flip for the AGB-diversity relationship from negative correlation to positive correlation. The regression analyses also showed that the AGB-diversity trend was less negative for the grazing grassland (T5) than for the grasslands under the other treatments. Moreover, the AGB-diversity trend under treatment T5 was observed positive if plant diversity was measured in terms of the Margalef and the Patrick richness indices. Complete details in the statistical characterization of the AGB-diversity trend are given in Appendix A. It is worth noting that a statistical significance was only associated with treatment T2.

## 3. Discussion

Previous research has found that, although environmental stresses adversely affected vegetation growth and wellbeing [29,30], grassland properties, and in particular, the species diversity in grasslands, were more affected by biotic disturbances because management practices usually led to more ubiquitous alterations to the plant communities [31]. For example, Niu et al. [32] found that, compared to the other species, the dominant species in a grassland system responded differently to disturbances, causing changes to the community composition and other properties and processes. In this study, the IV of the dominant species *Leymus chinensis* decreased to the lowest level under the grazing treatment, being, on average, 86.67%–94.29% lower than the values under the other treatments. Further analysis indicated that this may be related to the selective feeding behavior of the livestock, because animals preferentially graze on taller and more palatable plants [10]. As such, the growth of *Leymus chinensis* in the plant community was suppressed and its competence lowered, which jointly reduced its community dominance. From the perspective of functional groups, livestock grazing reduced the total IV of the *Poaceae* species, rendering the grass populations less competitive in the plant community, but allowing the more grazing- and trampling-resistant species (e.g., *Potentilla acaulis*) and species less palatable to livestock to gain a relative advantage. As a result, forb species increased significantly in the livestock grazing plot. In addition to the IVs, our data also revealed that the plots within the enclosure had the highest percentage of plant cover, exceeding the plot outside the enclosure by a margin of 16–37%. It is obvious that the enclosure eliminated grazing, allowing a smoother recovery of the degraded vegetation. Likewise, the number of perennial forbs in the community increased, whereas extra annual and biennial species were observed emerging, permitting the vegetation cover to reach sub-climax to climax levels over time. With regard to vegetation density, although the plants in plot T3 were significantly denser than in plots T2 and T5, plant density in T3 was only slightly higher than the other plots within the enclosure, showing that the perennial grass *Leymus chinensis*, in association with the herbaceous grass *Heteropappus altaicus* and the succulent grass *Orostachys fimbriatus,* maintained its dominance in the vegetation community (Appendix A).

Being one of the most frequently used quantitative ecosystem parameters, biomass is also a robust indicator of ecosystem stability and function [33,34,35]. In our study, five to eight years after livestock grazing was avoided, the highest AGB potential was achieved in the abandoned and degraded grasslands, which showed that the enclosure treatment ensured the aboveground parts of the plant to resume reproduction and propagation [36,37]. The lowest level of AGB was found in plot T5, which had been under constant grazing during the entire experiment, showcasing the strong, negative effects of grazing on grassland productivity that was in line with earlier studies [38,39]. Contrary to the AGB effect, grazing had significantly positive effects on plant diversity, evenness and richness indices. The reason was that grazing promoted species competition regarding the photoperiod by actively suppressing the tall, dominant species *Leymus chinensis* in the plant community [40]. It had been elaborated by Endara and Coley [41] that a change in the grassland community structure usually led to changes in the light transmission rate across the vertical vegetation layers. Evidence obtained from our study clearly illustrated that the shortest species in plot T5 benefitted from higher levels of light penetration following the removal of the dominant *Leymus chinensis* species. Another process that could have contributed to an increase in diversity is that the litter layer was removed by grazing. An important side effect of the litter removal by grazing was that the seeds of these shorter species became easier to come into contact with the soil and, therefore, had a higher rate of germination.

Studying the relationship between the productivity and species diversity of a grassland community is essential for understanding the structure and function of the grassland ecosystem [2,42]. Many studies have suggested that a positive correlation existed between species diversity and ecosystem productivity. Based on data obtained from a semi-arid grassland neighboring ours, for example, Zheng et al. [39] found that the grassland’s aboveground biomass was significantly positively associated with the Shannon–Wiener diversity index from 2005 to 2007. Evidence from an alpine steppe on the Qinghai–Tibet Plateau also showed a positive correlation of the species richness or diversity indices with both the above- and belowground biomass [43]. Contrary to these findings, our study showed consistently negative correlations between biomass productivity and the diversity indices on all experimental plots except T3. The same relationship turned positive on the annual mowing plot of T3, suggesting the mowing intensity triggered the sign switch for the diversity-productivity relationship, but further research is needed. Similar to our suggestion that mowing intensity may be a sign changer in diversity-productivity relationships, Wang [44] found, based on modeling evidence, that a change to the strength of many ecosystem processes had the potential to cause the diversity-productivity relationship to change sign. Moreover, in the same study that found a positive correlation above, Li et al. [43] also found that if the diversity-productivity relationship was evaluated based on Pielou’s evenness index, the sign of the relationship turned negative. They therefore proposed that spatial scale might be the sign changer for diversity-productivity relationships. What is more fundamental in the diversity-productivity relationship research is that a clear understanding of the control mechanisms of the complementarity or the selection effects among plant species is still lacking [45], and uncertainty remains in how to achieve a positive productivity response to diversity [46], especially during the recovery of degraded grasslands.

Grassland productivity is strongly related to the grassland’s long-term stability and resilience [47]. A high plant species diversity in a grassland community can meet the nutritional needs of the grazing livestock, resulting in better growth, higher reproduction and improved health [48]. Plot T1 in this study had the highest biomass value, but it had the lowest species diversity too, which is regarded as an unbalanced system from a systems perspective [49]. Although T5 under grazing management had increased species diversity, its sacrifice in livestock carrying capacity meant higher competition for grassland areas. Overall, plots T3 and T4 under mowing disturbances had suboptimal biomass and suboptimal species diversity in the same period of time. However, a balance between stability and performance was achieved in T3 and T4 and is, therefore, a recommendable management option in grassland conservation and sustainable use.

In conclusion, we found through field experiments that grassland biomass productivity was negatively associated with plant diversity. Although our data suggested that mowing intensity was a control over this productivity-diversity relationship, further research is still needed. We also found that mowing and enclosure were the best combination of management measures for the degraded meadow steppe to recover and produce, representing a progressive pathway to a more resilient grassland system based on current technologies.

## 4. Materials and Methods

### 4.1. Study Area

Field observations were conducted in the Xie’ertala Rangeland in the northern suburb of Hulunber City in Northeast Inner Mongolia, China (Figure 6). The elevation of the study area ranges between 600 and 660 m above sea level. The whole area sits in the transition zone from the Greater Khingan Mountains to the Mongolian Highlands. A temperate continental climate prevails in the region. The annual precipitation totals 350–400 mm, most of which falls between June and September. The annual temperature averages between −3 and 1 °C. The accumulative thermal units account for 1700–2300 degree-days per annum on the base temperature of 10 °C, which corresponds to a frost-free period of about 110 days. The dominant land use in the area is the *Leymus chinensis* meadow steppe, where a range of associated species including *Stipa baicalensis*, *Carex spp*, *Cleistogenes squarrosa*, *Poa sphondylodes*, *Achnatherum sibiricum*, etc., coexist. The dominant soils in the area are classified as Kastanozems [50].

### 4.2. Experimental Setting and Design

Experiments were conducted on a fence-enclosed field of 1500 m by 2500 m in size at Farm No. 12 of the Xie’ertala Rangeland (Figure 6). In June 2005, four treatments were implemented on the fence-enclosed field, which represented four management levels, namely, fallow (T1), zero mowing (T2), mowing once per year (T3) and mowing once per three years (T4). A plot outside the fence-enclosed field, which was open for livestock grazing, was also included in the experiment as the fifth treatment (T5). Grazing intensity on T5 was classified as intermediate. A summary of the characteristics of these treatments is given in Table 1.

### 4.3. Vegetation Survey

A vegetation survey was conducted in July 2013 when the plant growth was at the maximal level of the year using quadrats of 1 m × 1 m in size. Until 2013, T1 had been fence-enclosed for 5 years, and T2, T3 and T4 had been fence-enclosed for 8 years. Ten locations were randomly selected in each plot. Plant communities within the quadrat were surveyed to determine the coverage, height, density and aboveground biomass (AGB) for each plant species that appeared within the quadrat. The area under vegetation cover and the area of bare soil were visually estimated in situ by experienced field staff and the percent of vegetative area was derived as the coverage per plant species. Plant height was determined by the average height of five randomly selected plant individuals of the surveyed species. Plant density was measured by counting the total number of plant individuals per species. AGB was determined by collecting and weighing the aboveground part of the plant per species in each quadrat. Standing plants were cut at the soil surface per species and collected in sample bags. Fallen, withered parts on the soil surface were also collected. Plant samples were separated into green parts and dead parts in the laboratory. Fallen, withered parts were regarded as dead parts. Collected green and dead parts were weighed to record the fresh weight per sample. Dry weights were measured after the samples were oven-dried at 85 °C for 12 h.

### 4.4. Diversity Indices

The relative importance of a plant species in the plant community was evaluated using the importance value (IV), which is given by the following Equation (1) [51]:(1)$$IV=\frac{1}{4}\left(RC+RH+RD+RB_{a}\right)$$
where *RC* is the relative coverage of a plant species; *RH* is the relative height; *RD* is the relative density and *RB_a_* is the relative AGB. Here, the values of *RC*, *RH*, *RD* and *RB_a_* are defined as the percentage of a plant species’ value of the coverage, height, density and AGB to the sum of all plant species, respectively.

Plant diversity was evaluated using six indices, including the Shannon–Wiener diversity index [52], the Simpson diversity index [53], Pielou’s evenness index [54], Alatalo’s evenness index [55], the Margalef richness index [56] and the Patrick richness index [57]. These indices are defined by Equations (2) to (7), respectively. These indices are statistical representations of plant diversity in different aspects. The Shannon–Wiener and the Simpson indices reflect the species’ diversity within the plant community, Pielou’s and Alatalo’s indices measure the species distribution evenness across the community, and the Margalef and Patrick indices indicate the quantity of species of the plant community.
(2)$$H=-\sum \left(P_{i}\cdot \ln P_{i}\right)$$
(3)$$D=1-\sum \left(P_{i}^{2}\right)$$
(4)$$J=\frac{H}{\ln S}$$
(5)$$E_{a}=\frac{{\left(\sum P_{i}^{2}\right)}^{-1}-1}{e^{\sum P_{i}\cdot \ln P_{i}}-1}$$
(6)$$M=\frac{S-1}{\ln N}$$
(7)R=S
where *P_i_* is the relative importance value of plant species *i*, *S* is the total number of plant species sampled and *N* is the number of plant individuals sampled.

### 4.5. Data Analysis

Data were processed and analyzed using the R 4.1.3 software [58]. The analysis of variance (ANOVA) method coupled with the LSD test [59] was employed for multiple comparisons of diversity indices and biomass productivity against the field treatments.

## Figures and Tables

**Figure 1 plants-11-02037-f001:**
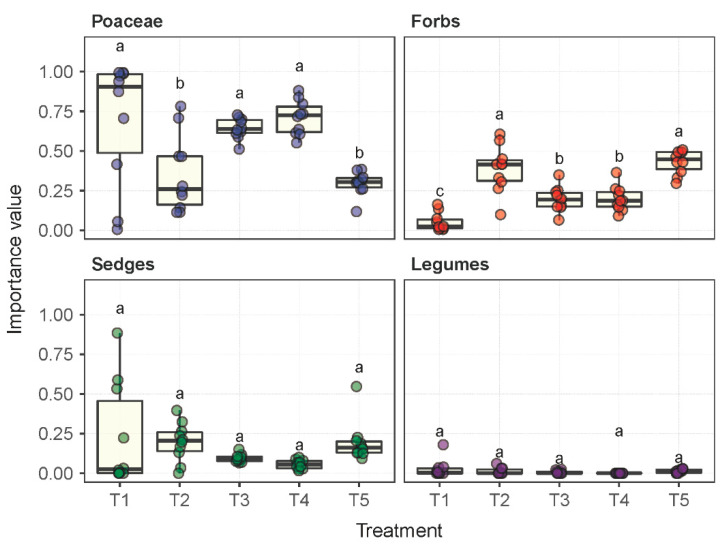
The importance values per plant species in each of the functional groups of *Poaceae*, forbs, sedges and legumes under the experimental treatments T1 through T5. Treatments with the same lowercase letter are not significantly different.

**Figure 2 plants-11-02037-f002:**
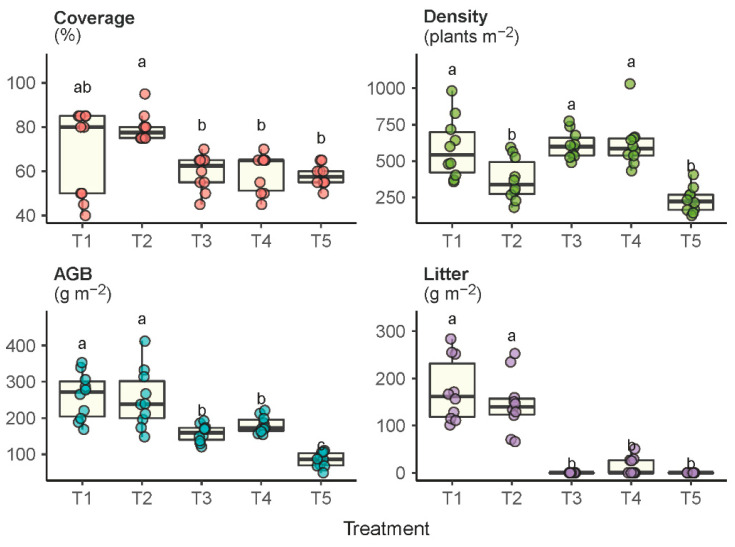
Plant community properties of coverage, density, aboveground biomass (AGB) and litter accumulation in response to the experimental treatments T1 through T5. Treatments with the same lowercase letter are not significantly different.

**Figure 3 plants-11-02037-f003:**
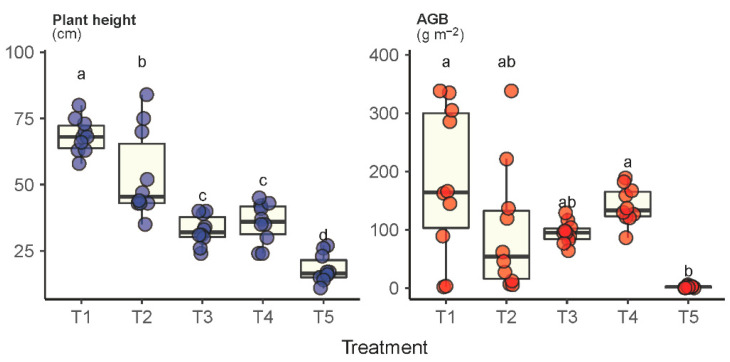
Plant height and aboveground biomass (AGB) of the dominant species *Leymus chinensis* under the experimental treatments T1 through T5. Treatments with the same lowercase letter are not significantly different.

**Figure 4 plants-11-02037-f004:**
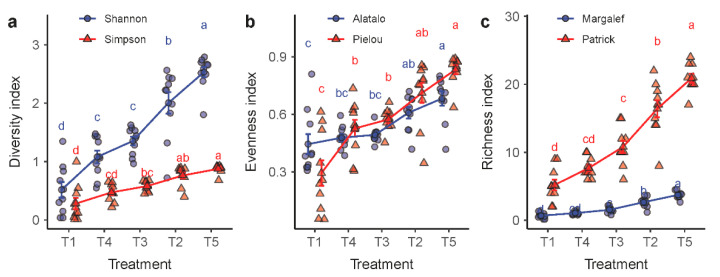
Variations in plant species diversity in response to the experimental treatments T1 through T5. Plant species diversity was measured in three groups of indices: (**a**) the Shannon–Wiener and Simpson’s diversity indices; (**b**) Alatalo’s and Pielou’s evenness indices; and (**c**) the Margalef and the Patrick richness indices. Treatments with the same lowercase letter are not significantly different.

**Figure 5 plants-11-02037-f005:**
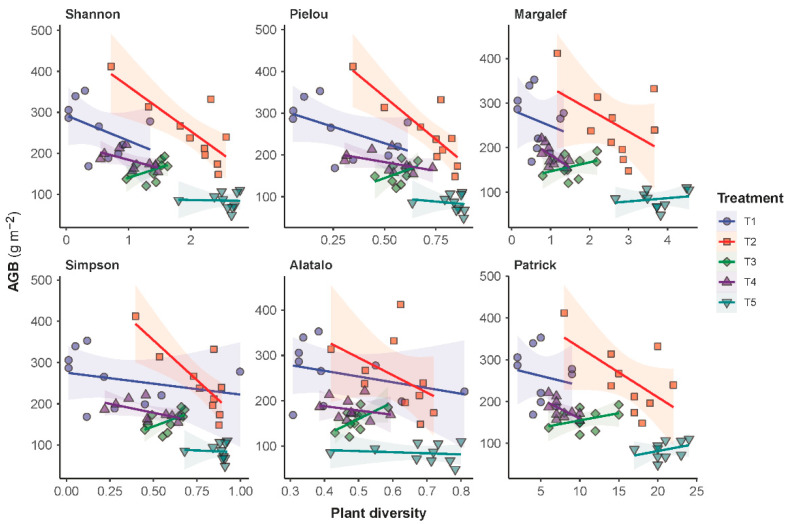
Relationships between the aboveground biomass (AGB) and the species diversity of the plant communities in the sampled grassland. Linear models were fitted to the AGB versus diversity data per experimental treatment. Parameters of the characterized models are given in Appendix A.

**Figure 6 plants-11-02037-f006:**
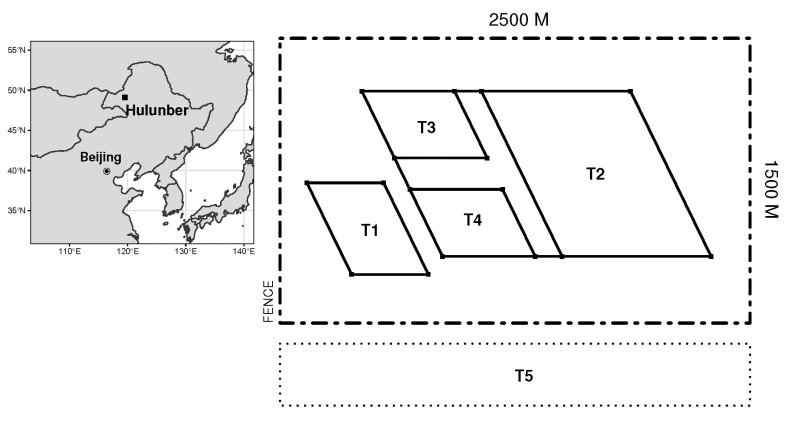
The study area and the plot arrangement in the field experiment in Hulunber. Plots T1 through T4 are located in a fence-enclosed field of 1500 m by 2500 m in size. Plot T5 is located on the open grassland outside the enclosure.

**Table 1 plants-11-02037-t001:** Description of the experimental treatments.

Plot	Treatment	Fence-Enclosed	Size (Hectares)	Description
T1	Fallow	Yes	13	Degraded forage grassland supplementally seeded with smooth bromegrass at 30 kg ha^−1^. In fallow since 2008.
T2	Fallow	Yes	46	Degraded forage grassland under natural restoration. Fence-enclosed since 2005.
T3	Mowing once per year	Yes	12	Degraded forage grassland. Fence-enclosed since 2005. Forage harvested by mowing in August every year at 7 cm above soil surface.
T4	Mowingonce per three years	Yes	12	Degraded forage grassland. Fence-enclosed since 2005. Forage harvested by mowing once every three years in August at 7 cm above soil.
T5	Livestock grazing	No	60	Degraded grazing grassland. Grazing intensity at 0.69 standard sheep units per hectare.

## Data Availability

The data analyzed in this study are either included in the paper and the Appendix A, or available by reasonable request to the corresponding author.

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
