# Peer review of "Enclosure in Combination with Mowing Simultaneously Promoted Grassland Biodiversity and Biomass Productivity"

_plants, 2022, doi:10.3390/plants11152037_

Round 1
Reviewer 1 Report
Please, check English language and cited literature, see in the text!

Reviewer 2 Report
Dear Authors,
You submitted a valuable research, with an interesting subject for the present environmental issues. Also, the presented data, even if there are quite old, from 2005-2013 research period, I found them to be relevant for nowadays.
Strong points: the article respects the journal's template and it's written in good English.
Now, let me emphasize some of the aspects observed after I read the entire paper.
Concerning the Methods, it's not clear how did you reached to the sizes mentioned in Table 1 (10 to 60 ha), since you mentioned before fence-enclosed fields of 1500-2500 m. There are m or sqm?
Also, at 4.4, please add explanation for each of the 6 used index, because it isn't clear what does it mean each of the 2-7 equations.
Suggestions:
1) Insert tables S1 and S2 into the main text.
2) Add some Conclusions, which can highlight some of the main results of your research.
Congratulations for your work and good luck!
